# Misconception about HIV/AIDS transmission among sexually active women in emerging regions of Ethiopia

Wubshet D. Negash[1,2*], Tadele Biresaw Belachew[1], Melak Jejaw[1],
Misganaw Guadie Tiruneh[1], Kaleb Assegid Demissie[1], Desale B. Asmamaw[3,4],
Elsa Awoke Fentie[3,5], Desalegn Anmut Bitew[3,6], Bewuketu Terefe[7],
Rahel Mulatie Anteneh[8], Lemlem Daniel Buffa[9], Tadesse Tarik Tamir[10],
Alebachew Ferede Zegeye[11]

**1** Department of Health Systems and Policy, Institute of Public Health, College of Medicine and Health Sciences, University of Gondar, Gondar, Ethiopia, **2** National Centre for Epidemiology and Population Health, The Australian National University, Canberra, Australia, **3** Department of Reproductive Health, Institute of Public Health, College of Medicine and Health Sciences, University of Gondar, Gondar, Ethiopia, **4** Monash Centre for Health Research and Implementation, Faculty of Medicine, Nursing and Health Sciences, Monash University, Melbourne, Australia, **5** School of population health, faculty of medicine and health science, University of New south wales, Sydney, Australia, **6** Kirby institute, faculty of medicine and health science, University of New south wales, Sydney, Australia, **7** Department of Community Health Nursing, School of Nursing, College of Medicine and Health Sciences, University of Gondar, Gondar, Ethiopia, **8** Depatment of Public Health, College of health science, Debre Tabor University, Debre Tabor, Ethiopia, **9** Department of Human Nutrition, Institute of Public Health, College of Medicine and Health sciences, University of Gondar, Gondar, Ethiopia, **10** Department of pediatrics and child health nursing, school of nursing, College of medicine and health sciences, University of Gondar, Gondar, Ethiopia, **11** Department of Medical Nursing, School of Nursing, College of Medicine and Health Sciences, University of Gondar, Gondar Ethiopia

* wubshetdn@gmail.com

## Abstract

### Background

Misconceptions about how HIV is transmitted persist among sexually active women, especially in emerging regions, and pose a barrier to effective prevention and control efforts. Therefore, this study aims to assess the magnitude of misconceptions about HIV transmission and its associated factors in emerging regions of Ethiopia.

### Methods

This study analyzed secondary data from a community based cross-sectional survey of 497 sexually active women in emerging regions of Ethiopia. Stata version 17.0 was used to analyze the data. Statistical analysis were completed after the data had been weighted. A binary logistic regression model was analyzed. Misconceptions about HIV transmission were addressed by examining whether healthy individuals can have HIV/AIDS and can contract HIV/AIDS through witchcraft, mosquito bites, or sharing AIDS-related foods. Odds ratio along with a 95% confidence interval (CI) was

**Data availability statement:** All relevant data are available within the manuscript and its Supporting information files.

**Funding:** The author(s) received no specific funding for this work.

**Competing interests:** The authors declare that they have no competing interests.

**Abbreviations:** AOR: Adjusted Odds Ratio; EA: Enumeration Area; EDHS: Ethiopian Demographic and Health Survey; DHS: Demographic Health Survey; HIV: Human Immunodeficiency Virus; PLWHA: People Living With HIV/AIDS.

generated to identify factors associated with misconceptions about HIV. A p-value less than 0.05 was declared as statistical significance.

## Results

Overall, 72.71% (95% CI: 68.62–76.46) sexually active women had misconceptions about HIV transmission. Those sexually active women from the poor household's class (AOR = 2.10; 95% CI: 1.06, 4.13) and those women who had no history of HIV test (AOR = 1.67 95% CI: 1.01, 2.75) were more likely to have misconception about HIV transmission.

## Conclusion

More than seven in ten sexually active women had misconceptions about HIV transmission. HIV prevention initiatives, such as voluntary HIV testing and post-test counseling, are needed to combat misconceptions about HIV transmission. Moreover, the administrative body of each region should facilitate extensive health education and campaigns to increase awareness of HIV transmission among sexually active women, particularly, those from poor households.

---

## Background

Human Immunodeficiency Virus and Acquired Immune Deficiency Syndrome (HIV/AIDS) continue to be a public health concern worldwide, with sub-Saharan Africa (SSA) bearing the highest burden [1]. Worldwide, over 76 million people have been living with Human Immunodeficiency Virus (HIV) [2]. Despite accounting for about 15.2% of the global population, Africans accounted for more than two-thirds of those cases, with 35 million infected, 15 million of whom have already died [3]. Over 38 million people globally are infected with HIV, with women accounting for more than half (19.2 million) of those infected [1,4]. In 2023, 39.9 million people around the world lived with HIV, 1.3 million people became newly infected with HIV, and 630, 000 people died from AIDS-related illnesses [5,6].

Globally, the annual number of fatalities from HIV/AIDS-related illnesses among people living with HIV (PLWH) fell from 1.7 million in 2004–770,000 in 2018. However, meeting the 2020 target of fewer than 500,000 deaths was not achieved [7,8]. East and Southern Africa, in particular, have 20.6 million PLWH and 800,000 newly infected in 2018 [5,6,9]. According to the 2016 Ethiopian demography and health survey (EDHS), just 39% of youth males and 24% of youth females have full HIV awareness [10]. From 2018 to 2030, over 360,000 deaths from HIV/AIDS-related illnesses are expected, and over 700 adolescents acquire HIV each day [11].

Good knowledge is key for protecting people from HIV infection, particularly among the most vulnerable groups [12]. Media exposure, wealth index, access to HIV voluntary counseling and testing (VCT), education and sex were significantly associated with HIV knowledge [13–15]. Despite active responses to HIV/AIDS by

various stakeholders, the 2019 statistics from USAID and WHO revealed that 37.9 million people were living with HIV/AIDS, 1.7 million people were newly infected with the disease, and 770,000 died from AIDS reasons [7–9]. Those sexually active women had at high risk of acquiring HIV/AIDS. Therefore, more efforts are needed to combat the disease [2,16].

The magnitude of misconceptions about HIV transmission is different across different parts of the globe. For example, according to a rural China survey, 70.4%, and 24% of women thought that HIV/AIDS is spread by mosquito bites, and handshakes, respectively [17]. In other contexts, like India, 85.3% and 79.3% of research participants, respectively, stated that sharing food and supernatural force may spread HIV/AIDS [18]. According to the 2016 Ethiopian Demographic and Health Survey, 70% of women think that eating certain foods and getting bitten by mosquitoes can spread HIV [19] and the overall magnitude of misconceptions about HIV/AIDS transmission in Ethiopia is 27.47% [20].

Despite Ethiopia's efforts to reduce HIV incidence, the country has not yet met its established targets [21,22]. In the emerging regions, progress has been particularly limited due to weaker health infrastructure and socio-cultural barriers. While there have been improvements in the availability and use of HIV prevention, care, and treatment services, misconceptions about HIV transmission remain widespread and under-studied in these areas. This study focuses on understanding such misconceptions among sexually active reproductive age women in emerging Ethiopian regions. Establishing this baseline is essential for designing culturally appropriate and effective interventions. Therefore, the aim of this study was to assess misconceptions about HIV transmission and its associated factors among sexually active women in emerging regions of Ethiopia.

## Methods

### Study settings and data source

The Ethiopian Demographic and Health Survey (EDHS) was used to study the health of women in emerging regions like Afar, Somalia, Benshangul Gumuz, and Gambela. These emerging regions are home to dispersed pastoralist and semi pastoralist people from life hostile poverty. In partnership with the Ethiopian Public Health Institute (EPHI) and the Federal Ministry of Health (FMoH), the Central Statistical Agency (CSA) conducted the survey from January 18 to June 27, 2016.

This study used the women's record (IR file) data set and extracted the outcome and its factors. Here is a link to the free data set that can be downloaded: https://dhsprogram.com/data/available-datasets.cfm. The EDHS employs a two-stage stratified sampling technique [23]. Prior to sample selection, proportional allocation was achieved in each stratum. As part of the first stage, 645 enumeration areas (EAs) were selected with a probability proportional to their number, and each stratum was randomly selected. Counting the number of households within each EA was done through household listing operations. Afterward, the household lists were used as sampling frames for selecting households. During the second phase, 28 households were selected with equal probability from each cluster. The 2016 Ethiopian Demographic and Health Survey (EDHS) initially selected 15,683 households nationwide, of which 4,680 were located in the four emerging regions (Afar, Benishangul-Gumuz, Gambela, and Somali). From these households, 3,850 women of reproductive age (15–49 years) who reported having had at least one sexual intercourse in their lifetime were identified as sexually active and eligible for inclusion. After excluding 633 women with missing data on key variables related to HIV transmission misconceptions, a total of 3,217 cases remained. To ensure representativeness and adjust for the complex survey design, sampling weights (v005/1,000,000) were applied. The final weighted sample size was 497 sexually active women from the emerging regions, which formed the basis for the analysis (S1 Data).

### Operational definition

The outcome variable was constructed by asking sexually active women (at least one sexual intercourse in their lifetime) of reproductive age four line item questions [24,25] as follows: (1) Can healthy looking persons have HIV/AIDS? (2) Can get AIDS by witchcraft/supernatural means? (3) Can get HIV/AIDS from mosquito bite? (4) Can get HIV/AIDS from sharing

foods from a person who had AIDS? All the questions had three response options as yes, no and do not know. Either a "no" answer for the first question or a "yes" answer for the second, third and fourth questions were considered as misconception and coded as 1. On the other hand, those who responded as "yes" for the first question and "no" and/or "do not know" for the rest of the questions were considered to have no misconception and coded as 0 [26–28].

Independent variables included in this study are listed in the table below based on their practical significance for misconception about HIV transmission (Table 1).

## Data processing and analysis

Stata version 14, software was used to analyze the data. In order to ensure that the EDHS sample was representative of the population and to obtain reliable estimations before data analysis, the dataset was weighted (v005/1000000) throughout the analysis. In this study, descriptive statistics and summary statistics were presented using frequencies, percentages, graphs, and tables. The data in EDHS may show more similarities between individuals in one cluster than in another cluster. However, the intra-cluster correlation coefficient (ICC) for the null model did not show significant variation in misconception between clusters. The result of the intra-cluster correlation coefficient is too small (0.012) from the null model. This 0.012 implies minimal between cluster variation in misconceptions about HIV transmission. Moreover, the data cannot be applicable to multilevel analysis. Furthermore, due to the relatively small sample size of sexually active women (n = 497) distributed across 645 clusters, the average number of individuals per cluster was insufficient to support reliable multilevel estimates. Therefore, we used a single-level binary logistic regression model. A bivariable analysis that calculated the proportion of misconceptions across the independent variables with their *p*-values was analysed. All the variables having a p-value less than 0.05 in bivariable analysis were used for multivariable analysis. For the multivariable analysis, adjusted odds ratio with 95% confidence intervals and a p-value of less than 0.05 were used to identify factors of misconception about HIV/AIDS. The results of the multivariable model was presented as adjusted odds ratio (AOR) while Variance inflation factor (VIF) was used to check for multicollinearity among independent variables and it was found that there was no multicollinearity **(mean value for the final model = 1.36).**

## Patient and public involvement statement

Sexually active women were included in this study by providing valuable information. Nevertheless, they have never been involved in the study design, protocol, data collection tools, and reporting disseminating the findings.

**Table 1. List of variables for the assessment of misconception about HIV/AIDS among sexually active women in emerging regions, in Ethiopia, 2016 (n = 497).**

| Variables | Description |
|---|---|
| Age | 15-24, 25–34, 35 and above |
| Place of residence | Rural, Urban |
| Women education Level | No formal education, Primary education, Secondary and higher education |
| Wealth index | Due to the high variability of observations from the original EDHS classification of households into five categories, the wealth index scores were re-categorized into three categories (poor, middle, and rich) by merging poorest with poorer and richest with richer for the ease of interpretation of principal component analysis [29,30]. |
| Media exposure | Those women who were either reading newspapers/magazine, or listening to radio and watching television at least once a week were considered as having media exposure whereas, those women who had neither read magazine/newspaper nor listen to radio/ television at all were considered as not having media exposure. |
| Current marital status | Married, Unmarried |
| Distance to the health facility | Big problem, Not big problem |

## Results

A total of 497 sexually active women participated in the study. The majority (62.17%) of the study participants were from households in the poor wealth index category. Of the study participants, 68.39% had no formal education. Half (50.62%) of the study participants perceived that distance to the health facility was a big problem (Table 2).

### Magnitude of misconception about HIV transmission

The overall, magnitude of misconception about HIV transmission among sexually active women in emerging regions was 72.71% (95% CI: 68.62, 76.46) (Fig 1).

### Factors associated with misconceptions about HIV transmission

In the multivariable logistic regression model, wealth index and history of HIV test were significantly associated with misconception about HIV transmission. Accordingly, those sexually active women from poor households had 2.1 times higher

**Table 2. Socio-demographic characteristics of sexually active women in emerging regions, in Ethiopia, 2016 (n = 497).**

| Variables | Categories | Frequency (n) | Percentage (%) |
|---|---|---|---|
| Age of respondents | 15-24 | 349 | 29.95 |
| | 25-34 | 195 | 39.26 |
| | 35 and above | 158 | 30.79 |
| Religion | Orthodox | 58 | 11.70 |
| | Muslim | 388 | 78.10 |
| | Protestant | 45 | 9.05 |
| | Other* | 6 | 1.05 |
| Household wealth index | | | |
| | Poor | 309 | 62.17 |
| | Middle | 42 | 8.37 |
| | Rich | 146 | 29.46 |
| Educational status of the participants | | | |
| | No formal education | 340 | 68.39 |
| | Primary | 137 | 27.49 |
| | Secondary & Higher | 20 | 4.12 |
| Occupation | | | |
| | Working | 198 | 39.88 |
| | Not working | 299 | 60.12 |
| Region | | | |
| | Afar | 102 | 20.47 |
| | Somalia | 247 | 49.72 |
| | Benshangul Gumuz | 114 | 22.97 |
| | Gambella | 34 | 6.85 |
| Residence | | | |
| | Urban | 113 | 22.82 |
| | Rural | 384 | 77.18 |
| Distance to the health facility | | | |
| | Big problem | 252 | 50.62 |
| | Not big problem | 245 | 49.38 |

*Traditional, no religion

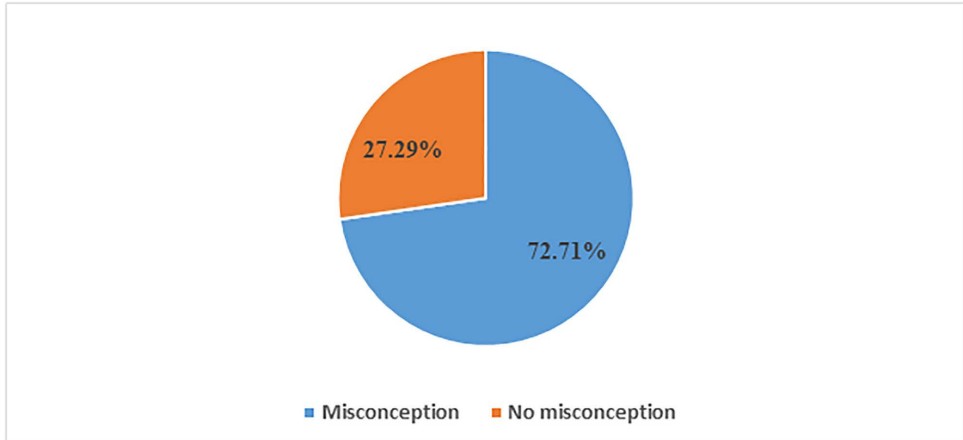

**Fig 1. Magnitude of misconceptions about HIV transmission in emerging regions, Ethiopia.**

odds of misconception about HIV transmission as compared with their counterparts (AOR = 2.10; 95% CI: 1.06, 4.13). The odds of misconception about HIV transmission was 1.67 times higher among those sexually active women who did not have history of HIV test (AOR = 1.67; 95% CI: 1.01, 2.75) as compared with those women who had history of HIV test (Table 3).

## Discussion

This study addresses a critical gap by providing a comprehensive assessment of misconceptions about HIV transmission among sexually active reproductive-age women specifically in the emerging regions of Ethiopia. These regions, characterized by unique socio-cultural and economic contexts (particularly pastoralist communities with lower economic status), have been historically under-researched regarding HIV knowledge and awareness. The findings establish a baseline understanding and is crucial for designing culturally appropriate and effective interventions to these emerging regions.

The findings of this study indicated that more than seven in ten, 72.71% (95% CI: 68.62, 76.46) sexually active women had misconception about HIV transmission in emerging regions of Ethiopia. These women believed that HIV can be transmitted from person to person by either supernatural means, mosquito bite or sharing food. On the other hand, only three in ten sexually active women had no misconception about HIV transmission at emerging regions of Ethiopia. The higher prevalence of misconceptions in this study compared to the national average (27.5%) may be explained by differences in the construction of the outcome variable. Our analysis included four misconception indicators, while the national estimate was based on only three. This study broader definition likely captured a wider range of misinformation and contributed to the higher prevalence observed among sexually active women in emerging regions [20].

Other findings in Malawi 56% [31], and Rwanda 46.4% [32] are lower than the current finding.

The discrepancy may be attributed to differences in study settings, definitions of misconception, and population characteristics. Although our study also included adolescents (15–24 years), it focused specifically on sexually active women in emerging, predominantly pastoralist regions, where access to HIV related information is limited and misconceptions may be more persistent. Socioeconomic and cultural factors in these communities such as lower education levels and weaker health infrastructure may contribute to higher prevalence.

Those sexually active women from the poor households had higher odds of misconception about HIV transmission as compared with their counterparts. The finding is similar with studies conducted in sub-Saharan Africa [33], and Ethiopia [34]. This might be wealthier respondents have a greater chance of getting health information and health education

**Table 3. Factors associated with misconceptions about HIV transmission among sexually active women in emerging regions, in Ethiopia, 2016 (n = 497).**

| Variables | Misconceptions about HIV | | COR (95% CI) | AOR (95% CI) | P-Value |
|---|---|---|---|---|---|
| | No | Yes | | | |
| **Age in years** | | | | | |
| 15-24 | 44(29.8) | 105(70.72) | 1 | 1 | |
| 25- 34 | 56(28.55) | 139(71.45) | 1.04(0.65, 1.65) | 0.82(0.47, 1.43) | 0.48 |
| 35 and above | 35(23.74) | 117(76.26) | 1.34(0.79, 2.22) | 0.98(0.53, 1.84) | 0.97 |
| **Religion** | | | | | |
| Orthodox | 33(56.75) | 25(43.25) | 1 | 1 | |
| Muslim | 81(20.80) | 307(79.2) | 4.99(2.81, 8.86) | 1.57(0.76, 3.35) | 0.22 |
| Other | 22(43.15) | 29(56.85) | 1.73(0.81, 3.69) | 1.1(0.46, 2.62) | 0.83 |
| **Educational status** | | | | | |
| No formal education | 65(19.00) | 275(81.0) | 11.10(4.06, 30.32) | 2.93(0.91, 2.44) | 0.07 |
| Primary education | 56(41.17) | 80(58.83) | 0.34(0.22, 0.51) | 0.68(0.39, 1.19) | 0.23 |
| Secondary and higher education | 15(72.25) | 6(27.75) | 1 | 1 | |
| **Current marital status** | | | | | |
| Married | 114(26.2) | 322(73.88) | 1 | 1 | |
| Unmarried | 21(34.88) | 40(65.12) | 0.66(0.37, 1.17) | 0.80(0.41, 1.56) | 0.50 |
| **Wealth index** | | | | | |
| Poor | 49(15.78) | 260(84.22) | 5.23(3.35, 8.17) | **2.10(1.06, 4.13)** | **0.03** |
| Middle | 14(34.55) | 27(65.45) | 0.35(0.17, 0.72) | 0.58(0. 27, 1.27) | |
| Rich | 72(49.50) | 74(50.50) | 1 | 1 | |
| **Media exposure** | | | | | |
| Yes | 64(45.93) | 75(55.07) | 0.30(0.19, 0.45) | 0.78(0.45, 1.37) | 0.36 |
| No | 72(20.08) | 286(79.92) | 1 | 1 | |
| **Perceived distance to the health facility** | | | | | |
| Not big problem | 81(33.10) | 164(66.90) | 0.56(0.37, 0.83) | 0.77(0.48, 1.24) | 0.29 |
| Big problem | 54(21.61) | 197(78.39) | 1 | 1 | |
| **Covered by health insurance** | | | | | |
| Yes | 2(63.72) | 1(36.28) | 0.26(0.17, 0.39) | 0.88(0.05, 14.01) | 0.93 |
| No | 134(27.11) | 360(72.89) | 1 | 1 | |
| **Ever been tested for HIV** | | | | | |
| Yes | 83(44.13) | 105(55.87) | 1 | 1 | |
| No | 53(17.03) | 256(82.97) | 3.84(2.55, 5.82) | **1.67(1.01, 2.75)** | **0.04** |
| **Place of residence** | | | | | |
| Rural | 81(21.12) | 303(78.88) | 0.35(2.22, 5.39) | 1.26(0.64, 2.51) | 0.50 |
| Urban | 55(48.14) | 59(51.86) | 1 | 1 | |
| **Region** | | | | | |
| Afar | 27(27.21) | 74(72.79) | 1 | 1 | |
| Somalia | 43(17.20) | 205(82.80) | 1.80(1.04, 3.11) | 1.39(0.74, 2.63) | 0.30 |
| Benshangul Gumuz | 49(43.22) | 65(56.78) | 0.49(0.28, 0.87) | 0.63(0.30, 1.31) | 0.22 |
| Gambella | 16(47.27) | 18(52.73) | 0.42(0.19, 0.93) | 1.04(0.37, 2.92) | 0.95 |

regarding HIV/AIDS transmission and related misconceptions [35,36]. Wealth enables more exposure to HIV transmission and to reject such misconceptions [36]. Therefore, increasing consistent and comprehensive knowledge about HIV/AIDS is recommended to minimize the knowledge gap between the poor and rich wealth rank households. The odds of

misconception about HIV transmission was higher among those sexually active women who had no history of HIV test as compared with those women who had history of HIV test. The finding is similar with studies conducted in Malawi [31], Ethiopia [37], and Uganda [38].

## Strengths and limitations

The study had the following limitations: Cross sectional nature of the data did not enable us to conclude the cause effect relationship of the findings. The study is the broad operational definition of sexually active women, which included all women of reproductive age who reported at least one sexual intercourse in their lifetime. This may have included women who were not currently sexually active, potentially influencing findings related to HIV transmission misconceptions and associated behaviors. The observed magnitude of HIV transmission misconceptions (72.71%) may appear higher than national figures, possibly due to how the variable was defined. In this study, any incorrect response to common transmission myths was classified as a misconception, which may have inflated the prevalence relative to stricter or alternative definitions.

The current finding of the study may change over time as a result of health facility expansion and community-based health insurance. It is also important to note that the distribution of the study participants is a weighted value, which means that some variables may not be considered in their actual value, and the sample size may not be large enough. Although EDHS data have a hierarchical structure, we used binary logistic regression due to a very low intra-cluster correlation (ICC = 0.012) and a small number of cases per cluster. As a result, potential community-level influences were not explored. The outcome was assessed by a few variables that existed in the EDHS. There may be other culture or norm related important variables used to assess misconceptions about HIV transmission. Despite the limitations, the finding can be generalized to the included regions.

## Conclusion

More than seven in ten sexually active women had misconception about HIV transmission. To overcome the problem of HIV misconceptions, comprehensive HIV interventions, such as HIV testing and counseling is needed at each health facility. Moreover, the government of each respective region needs to work extensively to increase the awareness of HIV transmission among sexually active women. Providers Initiated Testing and Counseling (PITC) of those people who attend healthcare services for different reasons is a very important intervention. The World Health Organization also recommends community engagement to HIV related knowledge [39]. Upon testing for HIV, people may be exposed to more comprehensive information about HIV/AIDS transmission. Counselors and healthcare providers at HIV testing facilities could use this evidence to influence decisions about capacity building and to ensure accurate information is conveyed to the clients.

## Supporting information

**S1 Data. CSV data used for the analysis.**
(CSV)

## Author contributions

**Conceptualization:** Wubshet Debebe Negash, Melak Jejaw, Desale B. Asmamaw, Desalegn Anmut Bitew, Bewuketu Terefe, Rahel Mulatie Anteneh, Tadesse Tarik Tamir, Alebachew Ferede Zegeye.

**Data curation:** Wubshet Debebe Negash, Tadele Biresaw Belachew, Melak Jejaw, Misganaw Guadie Tiruneh, Kaleb Assegid Demissie, Desale B. Asmamaw, Elsa Awoke Fentie, Desalegn Anmut Bitew, Bewuketu Terefe, Rahel Mulatie Anteneh, Tadesse Tarik Tamir, Alebachew Ferede Zegeye.

**Formal analysis:** Wubshet Debebe Negash, Tadele Biresaw Belachew, Melak Jejaw, Desale B. Asmamaw.

**Investigation:** Tadele Biresaw Belachew, Misganaw Guadie Tiruneh, Kaleb Assegid Demissie, Desale B. Asmamaw, Elsa Awoke Fentie.

**Methodology:** Tadele Biresaw Belachew, Melak Jejaw, Misganaw Guadie Tiruneh, Elsa Awoke Fentie, Bewuketu Terefe, Tadesse Tarik Tamir, Alebachew Ferede Zegeye.

**Resources:** Kaleb Assegid Demissie.

**Software:** Melak Jejaw, Misganaw Guadie Tiruneh, Kaleb Assegid Demissie, Elsa Awoke Fentie, Desalegn Anmut Bitew, Bewuketu Terefe, Lemlem Daniel Buffa.

**Supervision:** Desalegn Anmut Bitew, Tadesse Tarik Tamir.

**Validation:** Tadele Biresaw Belachew, Melak Jejaw, Misganaw Guadie Tiruneh, Kaleb Assegid Demissie.

**Visualization:** Kaleb Assegid Demissie, Rahel Mulatie Anteneh, Lemlem Daniel Buffa, Alebachew Ferede Zegeye.

**Writing – original draft:** Wubshet Debebe Negash.

**Writing – review & editing:** Wubshet Debebe Negash.

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
