## [Decision Letter · Decision Letter 0]

13 Aug 2024

Dear Dr. Negash,

Thank you for submitting your manuscript to PLOS ONE. After careful consideration, we feel that it has merit that needs some minor revisions before final publication. Therefore, we invite you to submit a revised version of the manuscript that addresses the points raised during the review process.

We look forward to receiving your revised manuscript.

Kind regards,

Oche Mansur Oche

Academic Editor

PLOS ONE

2. We notice that your supplementary figures are uploaded with the file type 'Figure'. Please amend the file type to 'Supporting Information'. Please ensure that each Supporting Information file has a legend listed in the manuscript after the references list.

Additional Editor Comments (if provided):

Reviewers' comments:

Reviewer's Responses to Questions

**Comments to the Author**

1. Is the manuscript technically sound, and do the data support the conclusions?

Reviewer #1: Yes

Reviewer #2: Yes

2. Has the statistical analysis been performed appropriately and rigorously?

Reviewer #1: Yes

Reviewer #2: Yes

3. Have the authors made all data underlying the findings in their manuscript fully available?

Reviewer #1: Yes

Reviewer #2: Yes

4. Is the manuscript presented in an intelligible fashion and written in standard English?

Reviewer #1: Yes

Reviewer #2: No

Reviewer #1: I have uploaded my review as an attachment. The author needs to review the title and compare it with the results of the study. Additionally, the author needs to reference UNAIDS/WHO recent report on the burden of HIV. There are several statements that also need to be referenced, this includes the questions that measured the misconception among the study participants. Furthermore, the author needs to the grammar, perhaps Grammarly could be helpful. Other comments will be found in the attached document.

Thank you.

Reviewer #2: The title requires a re-phrasing to capture of it being a study. Consider for example: "A Study of Misconception about HIV/AIDS among Sexually Active Women in Emerging Region of Ethiopia".

There are observed grammatical errors. Line 63: "were died" is incorrect. It should be simply "died"

Line 85 and 86 statement should be re-phrased to convey appropriate message meant to be passed across. Could be re-phrased as: "Despite the country's efforts at reducing the number of HIV cases, it is yet to achieve performance comparable to established objectives"

The sentence across Lines 116 and 117 needs to re-phrased for correctness of expression. Consider re-phrasing as: "The outcome variable was contructed by asking sexually active women (at least one sexual outcome in their lifetime) of reproductive age four line item questions

In list of variables table, the description for "Media Exposure", should entirely be changed to: "Those women who were either reading newspapers/magazine, or listening to radio and watching television at least once a week were considered as having media exposure whereas, those youths who had neither read magazine/newspaper nor listen to radio/ television at all were considered as not having media exposure".

If these corrections are made, the article is good to go!

**Do you want your identity to be public for this peer review?** For information about this choice, including consent withdrawal, please see our Privacy Policy

Reviewer #1: **Yes: ** Dr. Remi Abiola Oladigbolu

Reviewer #2: No

---

## [Author Response · Author response to Decision Letter 1]

20 Aug 2024

Authors’ response to editor and reviewer comments

We are very grateful to both the editor and reviewers for your comments and suggestions. All the concern raised so far will have an undeniable impact on improving the quality and readability of our scholarly work. Appreciating your effort and valuable comments, we have provided possible reflections and amended the raised concerns and questions. Kindly find our reflections here.

We hope you will consider the revised manuscript acceptable for publication in PLOSE ONE research journal.

S.no Editor comments Authors’ responses

1 Please ensure that your manuscript meets PLOS ONE's style requirements, including those for file naming. The PLOS ONE style templates can be found at

Dear editor thank you for the comment. We accepted the comments and revised it accordingly. Kindly check the revised manuscript.

2 We notice that your supplementary figures are uploaded with the file type 'Figure'. Please amend the file type to 'Supporting Information'. Please ensure that each Supporting Information file has a legend listed in the manuscript after the references list. We uploaded the figure as supporting information and added a legend at the end of the manuscript after the reference list. Kindly see the revised manuscript on page 19 line 365-367.

3 Please review your reference list to ensure that it is complete and correct. If you have cited papers that have been retracted, please include the rationale for doing so in the manuscript text, or remove these references and replace them with relevant current references. Any changes to the reference list should be mentioned in the rebuttal letter that accompanies your revised manuscript. If you need to cite a retracted article, indicate the article’s retracted status in the References list and also include a citation and full reference for the retraction notice. We tried to check all the references and we did not used retracted publications. Kindly see the references in the clean version of the manuscript on page 17-19 line -364.

Reviewer #1:

1 I have uploaded my review as an attachment. The author needs to review the title and compare it with the results of the study. Well received for the attachment. We tried to address the title and all your comments in the following section.

2 Additionally, the author needs to reference UNAIDS/WHO recent report on the burden of HIV. Dear editor thank you so much. We accepted the comment and referenced the recent UNAIDS/WHO reports. Kindly see page 4 line 69-71.

3 There are several statements that also need to be referenced, this includes the questions that measured the misconception among the study participants. Furthermore, the author needs to the grammar, perhaps Grammarly could be helpful. Other comments will be found in the attached document. We accepted the comment and added a reference for questions that measured misconceptions. Kindly see the revised manuscript on page 7 line 123-131.

4 Title

1 and 2 – you need to review the title of your work. It reads “Nearly seven in ten…”, however, your result showed over seven in ten (72.7%). Deadly sure. The result depicted more than seven in ten... we revised the title based on reviewer 2 suggestion. We corrected the result section accordingly. Kindly see the revised manuscript on page 1 line 1-2.

5 Abstract

57 – 58 – how will you increase the awareness of HIV transmission among the poor wealth class household? You need to specify your context: do you think awareness campaigns in these regions will help? Thank you for your question. We add the ways such as extensive health education and campaigns to increase the awareness of HIV transmission. Kindly see the revised manuscript on page 3 line 56-59.

6 Background

61 to 68 – Kindly reference the 2023 UNAIDS factsheet which is more accurate and recent. You need to review your grammar, perhaps you can use Grammarly to assist you. We accepted the comment and referenced the recent UNAIDS/WHO reports. Kindly see page 4 line 69-71. Moreover, we tried to revise the grammar of the whole manuscript.

7 69 to 73 – you should reference 2023 UNAIDS or WHO reports on the mortalities of HIV. This also applies to the data stated between 80 to 84. Dear editor thank you for the comment. We referenced UNAIDS and WHO. Kindly see the revised manuscript on Page 4 line 75.

8 91 to 94 – you should state the aim of the study before discussing the public health significance. Thank you for your suggestions. We stated the aim of the study before stating the public health significance. Kindly see the revised manuscript on page 5 line 93-97.

9 Study setting and data source

107 – Women’s record. Thank you for the editing. We accepted the edition and corrected in the manuscript. Kindly see on page 6 line 109.

10 110 to 114 – clearly state the two-stage stratified sampling technique.

Did you consider the power of your study when calculating a weighted sample of 497? Dear reviewer, we appreciate your comments. We tried to elaborate the two-stage stratified sampling technique in the revised manuscript on page 6-7 line 111-121. Regarding power of our study we mentioned in the limitations section on page 15 line 227-232.

11 Outcome variable

116 to 117 – Kindly reference the definition of a sexually active reproductive-age woman. Dear editor thank you so much for your question. We accepted the comment and added the references based on previous studies. Kindly refer page 7 line 123-124.

12 118 – “Yes” to “Health looking person can have HIV/AIDs” is not a misconception! Deadly sure. It is not a misconception. It was tying error. Kindly see on page 7 line 123-131.

13 118 to 121 – you need to reference the source of your questions on misconception. Referenced page 7 line 131.

14 Data processing and analysis

133 – summary statistics were presented using “text”? Please, be specific – mean, standard deviation for the quantitative variables, etc. We specified the expressions as frequencies and percentages. Kindly see the revised manuscript on page 8 line 141.

15 136 to 139 – the statements are not clear! Kindly review them. Revised on page 8-9 line 144-149.

Results

Factors associated with the misconception of HIV transmission

16 Table 1 – kindly reference the categorizations for your Wealth Index.

We accepted your recommendation and referenced on Page 8 table 1.

17 Table 2 – you have two variables on “Religion”! Dear editor thank you for your concerns. However, the first one is about religion whereas, the second one is about region. Kindly see table 2.

18 Discussions

197 to 199 – “The probability for high likelihoods of misconception among poorest household wealth categories is due to less or none frequent healthcare visit”. Did you obtain any information on the frequency of hospital visits for this category of participants? Perhaps you should provide another context. Dear editor thank you so much for your observation. We revised and added another context. Kindly see the revised manuscript on page 14 line 206-211.

19 205 – Do you mean Providers Initiated Testing and Counseling (PITC)? Sure. We want to say Providers Initiated Testing and Counseling (PITC). Kindly see the revised manuscript on page 15 line 215-216.

20 209 to 210 – kindly rephrase the statement. We appreciate your kind suggestions and rephrased on Page 15 line 220-221.

21 Strength and limitations

Could you have used a larger sample size? What was the power of this study? We stated in the limitation section. Kindly see on page 15 line 229-230.

22 Conclusion

Kindly look at the conclusion and compare it with your title. Thank you for your insight. We have seen the conclusion and amended the title.

Reviewer # 2

1 The title requires a re-phrasing to capture of it being a study. Consider for example: "A Study of Misconception about HIV/AIDS among Sexually Active Women in Emerging Region of Ethiopia". Thank you so much. We accepted and re-phrased the title as your suggestion. Kindly see page 1 line 1-2.

2 There are observed grammatical errors. Line 63: "were died" is incorrect. It should be simply "died" We accepted your comment and corrected it in the revised manuscript. Kindly see the clean version of the manuscript on page 4 line 64.

3 Line 85 and 86 statement should be re-phrased to convey appropriate message meant to be passed across. Could be re-phrased as: "Despite the country's efforts at reducing the number of HIV cases, it is yet to achieve performance comparable to established objectives" Dear reviewer, thank you for your comments. We accepted and re-phrased to convey the appropriate message. Kindly see the clean version of the manuscript on page 5 line 87-88.

4 The sentence across Lines 116 and 117 needs to re-phrased for correctness of expression. Consider re-phrasing as: "The outcome variable was contructed by asking sexually active women (at least one sexual outcome in their lifetime) of reproductive age four line item questions

Dear reviewer, thank you for your comments. We accepted and re-phrased to convey the appropriate message. Kindly see the clean version of the manuscript on page 7 line 123-125

5 In list of variables table, the description for "Media Exposure", should entirely be changed to: "Those women who were either reading newspapers/magazine, or listening to radio and watching television at least once a week were considered as having media exposure whereas, those youths who had neither read magazine/newspaper nor listen to radio/ television at all were considered as not having media exposure". Dear reviewer, thank you for your comments. We accepted and re-phrased to convey the appropriate message. Kindly see the clean version of the manuscript on page 8 table 1.

---

## [Decision Letter · Decision Letter 1]

18 Mar 2025

Dear Dr. Negash,

Thank you for submitting your manuscript to PLOS ONE. After careful consideration, we feel that it has merit but does not fully meet PLOS ONE’s publication criteria as it currently stands. Therefore, we invite you to submit a revised version of the manuscript that addresses the points raised during the review process.

We look forward to receiving your revised manuscript.

Kind regards,

Dereje Chala Chala Diriba, Ph.D.

Academic Editor

PLOS ONE

**Journal Requirements:**

**Additional Editor Comments:**

Dear Authors,

Thank you for submitting your article to PLOS ONE. I would lke to get address the following points before further process.

Title. A Study of Misconception about HIV/AIDS transmission among Sexually Active Women in Emerging Region of Ethiopia.

Is it possible to amend the title to ‘Misconception about HIV/AIDS Transmission among Sexually Active Women in Emerging Regions of Ethiopia’?

Abstract

Methods

Indicate the tool used to assess the misconception.

Background

More than 32 million people died as a result of the disease in the year 2019 [2]. Please use the most recent figure.

The misconception-related evidence should be reported.

Policymakers involved in implementing HIV prevention intervention programs will use our findings and valuable resources to combat HIV misconceptions.

Better to remove it from the manuscript.

Methods

The study area can be shortened more.

Ethiopian Demographic and Health Survey (EDHS) is also stated in the background section.

Remove ethics approval and informed consent from the manuscript.

The outcome measure is not indicated.

Results

Do you think the Religion frequency is correct?

What does Distance to the health facility Big problem 252 50.62 Not big problem 245 49.38 mean?

Why misconceptions are dichotomized as yes and no?

Who are poor, middle or rich? Please elaborate.

Please indicate the P value for each result.

Reviewer 2

Abstract

The definition and the gap of the subject under the study and the title of the research are not matching. As the focus of the study is on misconception, it will be better to describe the background under the abstract about misconception and study gaps related to it.

Better to be consistent when writing phrases and ideas. For instance: Can we use reproductive age women and sexually active women interchangeably? The authors stated…. “Reproductive age women”… in methods, but …”sexually active women…” in result part of the abstract.

Are all reproductive age women sexually active or are all sexually active women in a reproductive age?

Background

The authors mentioned about the magnitude and burden of the disease. However, the authors missed to introduce issues related to misconceptions, what types of misconceptions are there currently, evidences misconception and how it results in the raise of HIV/AIDs, magnitude/prevalence, as well as its burden. The authors should also have mentioned any studies previously conducted on misconceptions.

The authors are expected to rewrite the background more about misconception than about the magnitude and burden of the disease (HIV/AIDS) from global to Ethiopia, and clearly state the current gap which leads to misconception.

What is unique among those emerging regions? Do you have evidence that indicates the burden of HIV/AIDs among those emerging regions as compared to other regions? Better to give clue on these issues under the introduction part. What were the gaps of previously conducted studies and what was your plan to fulfil the gap?

Methods

What are those 645 enumeration areas? The stratification is by what and how many stratums were formed? The stratification was by what?

For the list of variables in table 1, “distance to the health facility was categorized as “big problem” and “not big problem”. When do we say it is a big problem and when do we say not a big problem? These things have to be mentioned as operational definition or under the outcome variables.

Results

The authors need to make the headings of the tables self-explanatory and give important information.

In table 2 for a variable religion, the category give “other” is exceeding an acceptable range or percept, and also what types of religions have been categorized under other should have been mentioned as footnote under the table.

The authors are not consistent in using their variable categories. For instance, in table 2 of their results they mentioned the categories for educational level as “No education”, “primary” and secondary and higher, however, in table 3, it was mentioned as “unable to read and write”, “primary”, and secondary and higher”. Again, in the table 1, it was mentioned as “No formal education”, “Primary education”, “Secondary and higher education”.

Discussion

The authors should have been discussed how the findings fulfilled the gaps, the importance of their studies. Rather than comparing the finding of the study with other studies done in somewhere, it is better to tell the reader on how their study fulfilled the gap of studies. That will be much more relevant than comparing the finding of the study with the findings of previously conducted studies.

The last paragraph of the discussion (line 215-223), has to be mentioned based on the findings and stated under the recommendation rather than under discussion.

Reviewers' comments:

Reviewer's Responses to Questions

**Comments to the Author**

Reviewer #1: All comments have been addressed

Reviewer #3: (No Response)

2. Is the manuscript technically sound, and do the data support the conclusions?

Reviewer #1: Yes

Reviewer #3: Partly

3. Has the statistical analysis been performed appropriately and rigorously?

Reviewer #1: Yes

Reviewer #3: No

4. Have the authors made all data underlying the findings in their manuscript fully available?

Reviewer #1: Yes

Reviewer #3: Yes

5. Is the manuscript presented in an intelligible fashion and written in standard English?

Reviewer #1: Yes

Reviewer #3: Yes

**Reviewer #1:**  I have reviewed the responses of the authors to my questions/comments/recommendations, and I am satisfied with their final submission.

**Reviewer #3:**  (No Response)

**Do you want your identity to be public for this peer review?** For information about this choice, including consent withdrawal, please see our Privacy Policy

Reviewer #1: **Yes: ** Remi Abiola Oladigbolu

Reviewer #3: No

---

## [Author Response · Author response to Decision Letter 2]

9 Apr 2025

Authors’ response to editor and reviewer comments

We are very grateful to both the editor and reviewers for your comments and suggestions. All the concern raised so far will have an undeniable impact on improving the quality and readability of our scholarly work. Appreciating your effort and valuable comments, we have provided possible reflections and amended the raised concerns and questions. Kindly find our reflections here.

We hope you will consider the revised manuscript acceptable for publication in PLOSE ONE research journal.

S.no Reviewer 1 Authors’ responses

1 Is it possible to amend the title to ‘Misconception about HIV/AIDS Transmission among Sexually Active Women in Emerging Regions of Ethiopia’? Thank you. We accepted and revised the title based on your suggestion. Kindly see page 1 line 1-2.

Abstract

2 Methods

Indicate the tool used to assess the misconception. Indicated. Kindly find page 3 line 47-50.

3 Background

More than 32 million people died as a result of the disease in the year 2019 [2]. Please use the most recent figure. Thank you for your comments. We accepted your comment and used the most recent figure. Page 4 line 66-67.

4 The misconception-related evidence should be reported. Reported in page 3 line 47-50.

5 Policymakers involved in implementing HIV prevention intervention programs will use our findings and valuable resources to combat HIV misconceptions. Better to remove it from the manuscript. We removed it. Page and line number not applicable

Methods

6 The study area can be shortened more. We shortened the study setting. Kindly see Page 7 line 114-115.

7 Remove ethics approval and informed consent from the manuscript. We removed it. Page and line number not applicable.

8 The outcome measure is not indicated. We tried to indicate the outcome measure. Page 3 line 47-50 and page 7-8 line 134-142.

9 Results

Do you think the Religion frequency is correct? Yes it is equal frequency with the other variables. We revised the categories. Kindly see page 10 table 2.

10 What does Distance to the health facility Big problem not problem mean? The distance to the health facility in DHS is categorized based on women's response to whether it is a big problem or not for the question “how do you perceive distance to the health facility?”

11 Why misconceptions are dichotomized as yes and no? We categorized the misconception based on other available literatures. This helps to compare, contrast and discuss with other existing findings. Page 8 line 142.

12 Please indicate the P value for each result. We added p value. Kindly see table 3 in the revised manuscript.

Reviewer #2:

1 Abstract

The definition and the gap of the subject under the study and the title of the research are not matching. As the focus of the study is on misconception, it will be better to describe the background under the abstract about misconception and study gaps related to it. We mentioned misconception in the abstract. Kindly see page 1 line 47-50 and page 5 line 89-96.

2 The authors stated…. “Reproductive age women”… in methods, but …”sexually active women…” in result part of the abstract.

We accepted your comments and made consistent. Kindly see the revised manuscript.

3 Are all reproductive age women sexually active or are all sexually active women in a reproductive age? All sexually active women are reproductive age women. We indicated in the manuscript on page 8 line 134-35.

4 Background

The authors mentioned about the magnitude and burden of the disease. However, the authors missed to introduce issues related to misconceptions, what types of misconceptions are there currently, evidences misconception and how it results in the raise of HIV/AIDs, magnitude/prevalence, as well as its burden.

The authors should also have mentioned any studies previously conducted on misconceptions. Thank you for your comment. We revised the background section. Kindly see on page 5, line 89-96.

5 The authors are expected to rewrite the background more about misconception than about the magnitude and burden of the disease (HIV/AIDS) from global to Ethiopia, and clearly state the current gap which leads to misconception. We revised the manuscript on page 5, line 89-96.

6 What is unique among those emerging regions? Do you have evidence that indicates the burden of HIV/AIDs among those emerging regions as compared to other regions? Better to give clue on these issues under the introduction part. What were the gaps of previously conducted studies and what was your plan to fulfil the gap? We revised the justification and stated in Page 5 line 100-106.

7 Methods

What are those 645 enumeration areas? The stratification is by what and how many stratums were formed? The stratification was by what? The Ethiopian Demographic and Health Survey (EDHS) employs a stratified, two-stage cluster sampling design.

1. The primary stratification variable is region, and the secondary stratification is residence. Within each region, EAs are further stratified by urban/rural residence. This creates two sub-strata within each region: urban EAs and rural EAs.

• Within each stratum, a specified number of EAs are selected using probability proportional to size (PPS) sampling. Based on the above approach a total of 645 EAs (202 urban and 443 rural) were used for the survey. The sampling frame for the EDHS is typically based on the most recent Population and Housing Census conducted by the Central Statistical Agency (CSA) of Ethiopia. This census provides a comprehensive listing of enumeration areas (EAs) across the country.

8 For the list of variables in table 1, “distance to the health facility was categorized as “big problem” and “not big problem”. When do we say it is a big problem and when do we say not a big problem? These things have to be mentioned as operational definition or under the outcome variables. The distance to the health facility in DHS is categorized based on women's response to whether it is a big problem or not for the question “how do you perceive distance to the health facility?”

We analyzed the available categorized data from the DHS.

9 Results

The authors need to make the headings of the tables self-explanatory and give important information.

We revised the heading of each tables. Kindly see table 1-3.

10 In table 2 for a variable religion, the category give “other” is exceeding an acceptable range or percept, and also what types of religions have been categorized under other should have been mentioned as footnote under the table We revised the variable religion. Kindly see table 2.

11 .The authors are not consistent in using their variable categories. For instance, in table 2 of their results they mentioned the categories for educational level as “No education”, “primary” and secondary and higher, however, in table 3, it was mentioned as “unable to read and write”, “primary”, and secondary and higher”. Again, in the table 1, it was mentioned as “No formal education”, “Primary education”, “Secondary and higher education”. Thank you for your insight. We revised the category for education. Kindly see the revised manuscript table 2.

11 Discussion

The authors should have been discussed how the findings fulfilled the gaps, the importance of their studies. Rather than comparing the finding of the study with other studies done in somewhere, it is better to tell the reader on how their study fulfilled the gap of studies. That will be much more relevant than comparing the finding of the study with the findings of previously conducted studies We accepted your comments and revised the discussion. Kindly see the revised manuscript on Page 14 line 191-197.

12 The last paragraph of the discussion (line 215-223), has to be mentioned based on the findings and stated under the recommendation rather than under discussion. We moved it to the recommendation section. Kindly see on page 15-16 line 235-240

---

## [Decision Letter · Decision Letter 2]

26 Jun 2025

Dear Dr. Negash,

Thank you for submitting your manuscript to PLOS ONE. After careful consideration, we feel that it has merit but does not fully meet PLOS ONE’s publication criteria as it currently stands. Therefore, we invite you to submit a revised version of the manuscript that addresses the points raised during the review process.

We look forward to receiving your revised manuscript.

Kind regards,

Dereje Chala Diriba, Ph.D.

Academic Editor

PLOS ONE

Journal Requirements:

**Additional Editor Comments:**

Please reconsider reporting guidelines and analysis.

Reviewers' comments:

Reviewer's Responses to Questions

**Comments to the Author**

Reviewer #3: All comments have been addressed

Reviewer #4: (No Response)

2. Is the manuscript technically sound, and do the data support the conclusions?

Reviewer #3: Yes

Reviewer #4: No

3. Has the statistical analysis been performed appropriately and rigorously?

Reviewer #3: Yes

Reviewer #4: No

4. Have the authors made all data underlying the findings in their manuscript fully available?

Reviewer #3: Yes

Reviewer #4: Yes

5. Is the manuscript presented in an intelligible fashion and written in standard English?

Reviewer #3: Yes

Reviewer #4: No

Reviewer #3: (No Response)

Reviewer #4: (No Response)

**Do you want your identity to be public for this peer review?** For information about this choice, including consent withdrawal, please see our Privacy Policy

Reviewer #3: **Yes: ** Adisu Ewunetu

Reviewer #4: No

---

## [Author Response · Author response to Decision Letter 3]

17 Jul 2025

Authors’ response to editor and reviewer comments

We are very grateful to both the editor and reviewers for your comments and suggestions. All the concern raised so far will have an undeniable impact on improving the quality and readability of our scholarly work. Appreciating your effort and valuable comments, we have provided possible reflections and amended the raised concerns and questions. Kindly find our reflections here.

S.no Editor comments Authors’ responses

Abstract

1 Background: What the authors defined and mentioned as a problem is the magnitude of HIV/AIDS, however, the title of the study is about misconception. Better to show that misconception is a problem since it is the subject under the study. Thank you for this insightful comment. We agree with the reviewer that the problem of misconception about HIV transmission should be more clearly highlighted in the Background section to align with the focus of the study. Accordingly, we have revised the background section of the abstract. Kindly see on page 3; line 41-44

2 Methods: Better to indicate the result of VIF, to indicate that there is no multi-collinearity. Thank you for the insightful comment. We have now assessed multicollinearity using the Variance Inflation Factor (VIF). Kindly see on page 9 line 164-166.

3 Better to include operational definition, and how some variables, for instance “misconceptions on HIV/AIDs”, “sexually active women” Thank you for the comment. We have now added operational definitions. Kindly see on page 7-8 line 131-145.

References

4 Some of the references are written in bold and others are not. So, better to be consistent and impartial across the document. Thank you for pointing this out. We have revised the reference formatting to ensure consistency throughout the document. All references are now uniformly formatted. Kindly see on page 18 line 280-396.

Reviewer 2

1 General comments:

I appreciated that the authors chose a good topic. However, the documents lack coherence, and needs critical language editions starting from the abstract to recommendation. Thank you for your constructive feedback. We have thoroughly revised the manuscript for coherence and clarity, from the abstract to the recommendations. Language editing has been applied throughout to improve readability and flow. Kindly see the whole revised manuscript.

Abstract:

2 Method is written as if the study was primarily conducted by the authors, which misleads someone who is interested to read the abstract only if get published. Thank you for the important observation. We have revised the Methods section to clearly reflect that the study was a secondary data analysis of the Ethiopian Demographic and Health Survey (EDHS), rather than a primary data collection. Kindly see on page 3 line 45-47.

3 Background:

Line 97-107: There are redundant statements within the same passage. Thank you for the feedback. We have revised lines 97–107 to remove redundancy and ensure clarity. Kindly see the revised manuscript on page 5 line 96-105.

Method:

4 Line 124-129: The numbers stated in the passage regarding the sample size, and how it was done is not clear. I think the authors need to be specific about the sample size their paper is concerned with. How did the authors came up with the last sample size (497)? Thank you for the insightful comment. We have revised the section to clarify the sampling process and how the final weighted sample size of 497 was determined. Specifically, we explained that:

A total of 15,683 households were initially included,

From which 3,850 sexually active women were identified,

After removing 633 cases with missing data, 3,217 women remained,

Finally, a weighted sample of 497 sexually active women from emerging regions was analyzed.

This stepwise clarification has now been explicitly stated in the revised Methods section. Kindly see the revised manuscript on page 6-7 line 122-130.

5 Outcome variable: How the study population was operationalized (sexually active women), is not convincing. How can we call a woman is sexually active when she had sex some years back (lifetime), not recently? Thank you for this important observation. We acknowledge that using “ever had sexual intercourse in a lifetime” as the operational definition of sexually active women may not accurately reflect current sexual activity. This broad definition follows the categorization used in the 2016 Ethiopian Demographic and Health Survey (EDHS), which limits our ability to refine the variable based on recency.

We have now clearly stated this limitation in both the Methods and Limitations sections, explaining that this operationalization may include women not currently sexually active, which could affect the associations observed with misconceptions about HIV transmission. Kindly see on page 15 line 231-236.

6 Data processing and analysis (Line 155): It is not clear, which enumeration areas does it refers to. It seems something is lacking. Thank you for pointing this out. We have provided additional detail on how EAs were selected using probability proportional to size sampling and their role as clusters in the survey design. This clarification now improves the transparency of the sampling and analysis process. Kindly see on page 6 line 114-122.

7 Obviously, EDHS data has a hierarchical nature and was already mentioned in the paper. However, the authors did not consider this in analysis, particularly, in the final analysis model. Thank you for this important comment. Although the EDHS data have a hierarchical structure, we conducted an intra-cluster correlation coefficient (ICC) analysis which showed minimal variation between clusters (ICC = 0.012). Additionally, the relatively small sample size per cluster limited the feasibility of multilevel modeling. Therefore, we used a single-level binary logistic regression model for the final analysis. This rationale has now been clearly explained in the Methods and limitations section. Kindly see the revised manuscript on Page 8-9 line 150-159 and page 16 line 242-246.

8 Results:

The magnitude of misconception seems exaggerated when compared to the national average. It needs to be checked whether how the outcome variable was generated impacted this. Thank you for this valuable observation. We agree that the higher magnitude of misconceptions in our study compared to the national average may be influenced by differences in how the outcome variable was constructed. Our study used a broader definition, including four indicators of misconceptions, while the national estimate was based on fewer items. We have clarified this in the discussion section to explain that the wider definition likely captured more misconceptions, contributing to the higher prevalence observed. Kindly see on page 14 line 201-210.

Discussions:

9 Line 206-207: The authors justified that the discrepancy could be due to the fact other studies are conducted among adolescents. However, the study also included adolescents (15-24years). This seems confusing and need to be discussed in another different way. Thank you for the helpful comment. We agree that the initial explanation was unclear. We have revised this section to clarify. Kindly see the revised manuscript on Page 14 line 213-215.

10 Overall, the paper needs to be critically revised as it currently stands. Thank you for your overall feedback. We have carefully revised the entire manuscript to improve clarity, coherence, and academic rigor. Language editing has been applied throughout to enhance readability and flow. We believe these revisions have significantly strengthened the paper. Kindly see the whole revised manuscript.

---

## [Decision Letter · Decision Letter 3]

18 Aug 2025

Misconception about HIV/AIDS Transmission among Sexually Active Women in Emerging Regions of Ethiopia.

PONE-D-24-19228R3

Dear Dr. Negash,

We’re pleased to inform you that your manuscript has been judged scientifically suitable for publication and will be formally accepted for publication once it meets all outstanding technical requirements.

Kind regards,

Dereje Chala Diriba, Ph.D.

Academic Editor

PLOS ONE

Additional Editor Comments (optional):

Dear authors,

You have addressed all comments from reviewers and editor. Please check affiliation of authors are written correctly as a noun. 

Reviewers' comments:

Reviewer's Responses to Questions

**Comments to the Author**

Reviewer #3: All comments have been addressed

2. Is the manuscript technically sound, and do the data support the conclusions?

Reviewer #3: Yes

3. Has the statistical analysis been performed appropriately and rigorously?

Reviewer #3: Yes

4. Have the authors made all data underlying the findings in their manuscript fully available?

Reviewer #3: Yes

5. Is the manuscript presented in an intelligible fashion and written in standard English?

Reviewer #3: Yes

Reviewer #3: (No Response)

**Do you want your identity to be public for this peer review?** For information about this choice, including consent withdrawal, please see our Privacy Policy

Reviewer #3: **Yes**

---

## [Editor Report · Acceptance letter]

PONE-D-24-19228R3

PLOS ONE

Dear Dr. Negash,

I'm pleased to inform you that your manuscript has been deemed suitable for publication in PLOS ONE. Congratulations! Your manuscript is now being handed over to our production team.

Kind regards,

on behalf of

Dr. Dereje Chala Chala Diriba

Academic Editor

PLOS ONE